# Local Recurrence following Radiological Complete Response in Patients Treated with Subsegmental Balloon-Occluded Transcatheter Arterial Chemoembolization for Hepatocellular Carcinoma

**DOI:** 10.3390/cancers15204991

**Published:** 2023-10-14

**Authors:** Dong Il Gwon, Gun Ha Kim, Hee Ho Chu, Jin Hyoung Kim, Gi-Young Ko, Hyun-Ki Yoon

**Affiliations:** Department of Radiology and Research Institute of Radiology, University of Ulsan College of Medicine, Asan Medical Center, Seoul 05505, Republic of Korea; kimgh.rad@amc.seoul.kr (G.H.K.); d180717@amc.seoul.kr (H.H.C.); khkimrad@amc.seoul.kr (J.H.K.); kogy@amc.seoul.kr (G.-Y.K.); hkyoon@amc.seoul.kr (H.-K.Y.)

**Keywords:** hepatocellular carcinoma, transcatheter arterial chemoembolization, balloon, recurrence

## Abstract

**Simple Summary:**

Balloon-occluded transcatheter arterial chemoembolization (B-TACE), in which a microballoon catheter is used to temporarily occlude the tumor feeding artery, was introduced to increase complete response (CR) rates. However, assessing the quality of CR in terms of the therapeutic outcomes of B-TACE is necessary because the evaluation of local recurrence (LR) may provide more detailed information on the ability of the B-TACE to result in a local cure. This study aimed to determine the LR rate and identify factors associated with LR in patients who achieve a radiological CR after undergoing subsegmental B-TACE for HCC. The CR rate after B-TACE was 97.2% at first follow-up. Oily subsegmentectomy, defined as radiological CR of the HCC and peritumoral parenchymal necrosis, can be considered as an index of successful treatment because it did not demonstrate any LR.

**Abstract:**

The aim of this study was to determine the local recurrence (LR) rate and identify factors associated with LR in patients who achieve a radiological complete response (CR) after undergoing balloon-occluded transcatheter arterial chemoembolization (B-TACE) for hepatocellular carcinoma (HCC). From November 2017 to September 2021, 60 patients (44 men, 16 women; mean age, 63.5 years; range, 39–82 years) with 72 HCCs (mean diameter, 31 mm; range, 10–50 mm) who underwent subsegmental B-TACE were included in this retrospective study. Radiological and clinical evaluation of oily subsegmentectomy, defined as radiological CR of the HCC and peritumoral parenchymal necrosis, was performed. The CR rate was 97.2% (70 of 72 HCCs) at first follow-up (mean, 41 days; range, 14–110 days). Overall, 13 HCCs (19.7%) demonstrated LR at a mean of 29.8 months (range, 3–63 months) and cumulative LR rates were 1.5% 14.2% 21%, 21%, and 21% at 6, 12, 24, 36, and 48 months, respectively. In 28 (38.9%) of 72 HCCs, oily subsegmentectomy was achieved, tumor markers were normalized, and LR did not occur. The oily subsegmentectomy-positive group had a significantly lower LR rate than the oily subsegmentectomy-negative group (*p* = 0.001). Age ≥65 years (adjusted hazard ration (HR), 0.124; 95% confidence interval (CI), 0.037–0.412; *p* < 0.001) and peripheral location (adjusted HR, 0.112; 95% CI, 0.046–0.272; *p* < 0.001) were independent predictive factors of LR. Subsegmental B-TACE can be an effective method with a high initial CR rate and low LR incidence. Oily subsegmentectomy can be considered as an index of successful treatment because it did not demonstrate any LR.

## 1. Introduction

According to the Barcelona Clinic Liver Cancer (BCLC) staging system, hepatocellular carcinoma (HCC) classified as very early-stage (<2 cm; BCLC 0) and early-stage disease (BCLC A) nodules has recommended curative treatments such as radiofrequency ablation (RFA), surgical resection (SR), and liver transplantation (LT) [1,2,3]. However, in cases in which all of these treatments are unfeasible, transcatheter arterial chemoembolization (TACE) can be an alternative treatment.

The complete response (CR) of HCC at the first TACE is the most robust predictor of favorable outcomes in patients with HCC [4]. Even if CR is obtained after successful TACE, local recurrence (LR) may occur [5,6,7,8,9]. Using the modified Response Evaluation Criteria in Solid Tumors (m-RECIST) in post-TACE evaluation tends to overestimate radiologic results, and a discrepancy between the absence of radiological viability and actual pathological necrosis commonly occurs [10,11,12].

Balloon-occluded transcatheter arterial chemoembolization (B-TACE), in which a microballoon catheter is used to temporarily occlude the tumor feeding artery, was introduced to increase CR rates [13,14]. Although no definitive indications for B-TACE have been established to date, several studies have reported that initial tumor responses are better among patients with HCC who undergo B-TACE than other TACE techniques, including conventional TACE (C-TACE) and drug-eluting bead TACE [15,16,17,18,19,20,21]. However, assessing the quality of CR in terms of the therapeutic outcomes of B-TACE is necessary because the evaluation of LR may provide more detailed information on the ability of the B-TACE to result in a local cure. Therefore, this study aimed to determine the LR rate and identify factors associated with LR in patients who achieve a radiological CR after undergoing subsegmental B-TACE for HCC.

## 2. Materials and Methods

Altogether, 60 patients (44 men, 16 women; mean age, 63.5 years; range, 39–82 years) with 72 HCCs (mean diameter, 31 mm; range, 10–50 mm) who underwent subsegmental B-TACE from November 2017 to September 2021 were included in this study. Study patients presented inaugural HCCs (n = 46) or newly diagnosed HCCs (n = 14) after RFA (n = 5) or SR (n = 9) on different lesions. Patients with unresectable HCC(s) not eligible for curative treatments (including RFA, SR, and LT) with at least one lesion, along with patients on the transplantation waiting list, were included. Patients were excluded if they had undergone TACE previously, had a concurrent malignancy other than HCC, had >5 HCCs, or had an HCC sized <1 cm or >5 cm. The baseline characteristics of the study patients are summarized in Table 1.

B-TACE was performed by one of three interventional radiologists (experience: 6, 15, and 20 years). Before B-TACE, superior mesenteric and common hepatic arteriographies were performed using a 5-F catheter (Rösch hepatic catheter; Cook, Bloomington, IN) to assess portal flow direction, anatomy of the hepatic artery, and feeding arteries, along with tumor blush and location. For subsegmental B-TACE, a 2.0-F microballoon catheter (Optimo PB, Tokai Medical Products, Yokohama, Japan) was advanced as close as possible to the tumor feeding artery. When selective catheterization could not be performed because of tortuosity or small size, a microballoon catheter was placed as close as possible to the tumor feeding artery. After the appropriate placement, a microballoon was generally inflated to the diameter of the target artery and then chemoinfusion was performed under the microballoon occlusion, as per the standard protocol of our institution. For chemoinfusion, 0.5 mg/mL cisplatin dissolved in distilled water was infused for 15 min, with the cisplatin dose being 2 mg/kg (4 mL/kg) per patient weight. Iodized oil (Lipiodol; Guerbet, Roissy, France)/cisplatin emulsion was then infused. The ratio of cisplatin to iodized oil was 1:2. In most patients, cisplatin-based TACE was performed, while in patients with an allergy to cisplatin or renal dysfunction, doxorubicin-based TACE was performed (dose; 25–50 mg). The endpoint of infusion was the complete filling of iodized oil in the tumor as well as overflow into the intrahepatic collateral arteries or peritumoral portal veins. Subsequently, an embolic agent, 100–300 µm gelatin particles (Nexsphere: Nextbiomedical, Incheon, Republic of Korea), was injected. The endpoint of embolization was after densely filling the tumor feeding artery with gelfoam slurry and maintaining stasis after microballoon deflation. When the gelfoam slurry flowed beyond the catheter tip, the slurry was promptly injected under the deflated microballoon until it densely filled the tumor feeding branches.

The patients were closely monitored for at least 3 days to detect and manage any adverse events or post-embolization syndrome after B-TACE. At the first follow-up visit, the subsequent management plan was decided by multidisciplinary teams depending on the patient’s general condition, laboratory findings, and tumor response evaluation on follow-up computed tomography (CT).

The study endpoints were the technical success, complications, and effectiveness of B-TACE, along with identification of LR. Technical success was defined as the successful placement of the microballoon catheter, along with the injection of chemotherapeutic and embolic agents to the target HCC. Complications were classified as major or minor, according to the Society of Interventional Radiology clinical practice guidelines [22]. Major complications were defined as those necessitating additional treatment, including admission to a hospital for therapy, an unplanned increase in the level of care, prolonged hospitalization (>48 h), permanent adverse sequelae, or death. All other complications were considered minor. Post-embolization syndrome, defined as transient fever, abdominal pain, nausea, and/or vomiting, was not considered a posttreatment morbidity. However, fever that persisted for >3 days was regarded as a major complication. Effectiveness, defined as the initial radiologic response, was evaluated every 4–6 weeks by dynamic contrast-enhanced abdomen CT, according to the m-RECIST criteria. Radiologic response was defined as CR or partial response (PR) on the first follow-up CT. At the first follow-up CT, the Hounsfield units (HU) of regions demonstrating iodized oil accumulation were quantitatively measured by hand; a region of interest was drawn along the tumor periphery on the axial image containing the maximum tumor diameter, using a pre-B-TACE CT examination as a reference. These measurements were performed at a specific window (50 HU) and width (350 HU) [7]. Time to LR was measured from the date of B-TACE to the date of the follow-up CT scan at which a viable tumor around the B-TACE-treated HCC was first observed. Oily subsegmentectomy was defined as the radiological CR of the HCC and peritumoral parenchymal necrosis of any thickness. Patient survival was defined as the time interval between the B-TACE and death or last follow-up. Post-recurrence survival (PRS) was defined as the time interval between the detection of recurrence on CT and death or last follow-up.

Statistical analyses were performed using the SPSS 21.0 software and R software version 3.6.1 (R Foundation for Statistical Computing, Vienna, Austria). Patient survival and PRS rates were calculated according to the Kaplan–Meier method. Two-sided *p* < 0.05 was considered significant. Continuous variables are expressed as means ± standard deviations (SDs) and analyzed using Student’s *t*-test. Categorical variables are expressed as numbers (%) and analyzed using the chi-square test or Fisher’s exact test, as appropriate. Univariate and multiple Cox proportional hazard regression analyses clustered by patients were performed to determine the predictors of LR. Variables with *p* < 0.2 in the univariate analysis were entered into the multiple analysis. In case of zero-inflated data, the log-rank test was used. Cumulative LR rates were determined using the Kaplan–Meier method.

## 3. Results

### 3.1. B-TACE Procedure and Complications

The subsegmental B-TACE procedures achieved technical success in all 60 patients (Figure 1). Cisplatin-based B-TACE was performed in 57 patients, and doxorubicin-based B-TACE was performed in three. The embolization covered one subsegment in 42 patients, two subsegments in 16, and three subsegments in two. In three patients, extrahepatic collateral vessels (two right inferior phrenic artery and one right gastroepiploic artery) were observed and embolized with lipiodol emulsion and gelatin particles.

After the procedure, 34 (56.7%) patients experienced post-embolization syndrome, of whom 7 required a prolonged hospital stay because of persistent fever for >3 days; these were considered major complications. Asymptomatic ischemic cholangiopathy, considered a minor complication, was incidentally detected in one patient on follow-up CT. Therefore, the major and minor complication rates were 11.7% (7 of 60 patients) and 1.7% (1 of 60 patients), respectively. Aspartate aminotransferase (AST) (median ± SD, 36 ± 18 IU/L to 379 ± 293 IU/L; *p* < 0.001), alanine aminotransferase (ALT) (median ± SD, 33 ± 25 IU/L to 432 ± 414 IU/L; *p* < 0.001), and total bilirubin (median ± SD, 0.8 ± 0.4 mg/dL to 1.7 ± 0.8 mg/dL; *p* < 0.001) levels were significantly increased within 3 days after B-TACE. All biochemical markers (AST, ALT, and total bilirubin) returned to baseline levels within 6 weeks after B-TACE.

### 3.2. Follow-Up

The treatment outcomes are presented in Figure 2. A first follow-up CT was available for all patients (mean, 40 ± 14.8 days; range, 17–63 days). According to the m-RECIST criteria, 70 (97.2%) HCCs demonstrated CR, whereas two (2.8%) had PR. Therefore, all HCCs achieved a radiologic response on the first follow-up CT. The mean HU was 563 ± 208 (range, 122–960). At the first follow-up, alpha-fetoprotein (AFP) and protein induced by vitamin K absence II (PIVKA-II) were present in 56 patients and 48 patients, respectively. AFP (median ± SD, 829 ± 3313 ng/mL to 40 ± 193 ng/mL; *p* = 0.065) did not show a significant decrease after B-TACE, whereas PIVKA-II levels (median ± SD, 371 ± 986 mAU/mL to 30 ± 15 mAU/mL; *p* = 0.021) decreased significantly.

Among the 70 HCCs with a CR, an oily subsegmentectomy confined within 3 cm from the HCC was observed in 28 (38.9%) HCCs. The mean HU of lipiodol-laden HCCs was 592 ± 1888 (range, 243–980) (Figure 1e). At the first follow-up, the tumor marker levels were normalized in these 28 patients with oily subsegmentectomy. Oily subsegmentectomy was significantly associated with the peripherally located HCC (Fisher’s exact test, *p* = 0.008).

### 3.3. Local Recurrence

Among the 58 patients (70 HCCs) who achieved radiological CR, two patients (with one HCC each) who underwent LT were excluded, and two (with one HCC each) others were not followed up. Clinical follow-up information until patient death or the end of the study was available for the remaining 54 patients with 66 HCCs. The cutoff date for data analysis was 30 April 2023. During a mean follow-up of 42.8 months (range, 8.1–65.7 months), a total of 13 HCCs (19.7%) demonstrated LR with a mean of 29.8 months (range, 3–63 months). The cumulative LR rates were 1.5%, 12.7%, 21.6%, 21.6%, and 21.6% at 6, 12, 24, 36, and 48 months, respectively (Figure 3). None of the 28 HCCs with oily subsegmentectomy demonstrated any LR (Figure 4). The univariate Cox proportional hazard regression analyses revealed that age ≥65 years (adjusted hazard ratio (HR), 0.243; 95% confidence internal (CI), 0.074–0.804; *p* = 0.021), maximum HCC size (>3 cm) (adjusted HR, 0.475; 95% CI, 0.155–1.454; *p* = 0.192), and peripheral location (adjusted HR, 0.319; 95% CI, 0.101–1.003; *p* = 0.051) were associated with the LR rate. The hazard ratios of two variables (Child–Pugh B and oily subsegmentectomy) were inestimable in the univariate Cox proportional hazard regression analysis since none of the HCCs with these variables indicated LR. The univariate log-rank analysis revealed that the oily subsegmentectomy-positive group had a significantly lower LR rate than the oily subsegmentectomy-negative group (*p* = 0.001) (Figure 5a). In addition, the multiple Cox proportional hazard regression analysis revealed that age ≥65 years (adjusted HR, 0.124; 95% CI, 0.037–0.412; *p* < 0.001) and peripheral location (adjusted HR, 0.112; 95% CI, 0.046–0.272; *p* < 0.001) were independent predictive factors for LR (Figure 5b,c) (Table 2).

All 13 patients with an LR were suitable for a second conventional TACE treatment.

### 3.4. Overall Survival and Post-Recurrence Survival

Following B-TACE, 12 (22.2%) of 54 patients died, whereas 42 (77.8%) remained alive. The causes of death were liver failure in 7 (58.3%) patients, tumor progression in 3 (25%), and events other than liver decompensation or tumor progression in 2 (16.7%). The cumulative survival rates were 100%, 94.4%, 88.9%, 84.9%, and 78.5% at 6, 12, 24, 36, and 48 months, respectively.

Among 13 patients with an LR, 6 (46.2%) patients died, whereas 7 (53.8%) remained alive. The causes of death were liver failure in 4 (66.7%) patients and tumor progression in 2 (33.3%). The median PRS rate was 35.9 months (range, 29–43 months) and the cumulative PRS rates were 92.3%, 84.6%, 76.9%, 44.8%, and 44.8% at 6, 12, 24, 36, and 48 months, respectively.

## 4. Discussion

In the present study, radiological CR was achieved in 72 (97.3%) HCCs and PR in two (2.7%) HCCs after subsegmental B-TACE. During the follow-up (mean, 42.8 months; range, 8.1–65.7 months), CR was maintained in 53 (79.4%) HCCs, whereas 13 (20.6%) demonstrated LR at a mean of 29.8 months (range, 3–63 months). The cumulative LR rates were 1.5%, 14.2%, 21%, 21%, and 21% at 6, 12, 24, 36, and 48 months, respectively. Previous studies have reported on some cases in which LR occurred despite achieving radiological CR, with an incidence of 11.1%–53% [8,9,10,11,12]. Shirono et al. reported that the LR period for B-TACE was longer than that for C-TACE and DEB-TACE, even when CR was achieved. Therefore, subsegmental B-TACE can be an important method for maintaining a sustained CR and prolonging the time of recurrence. Moreover, it could reduce the need for retreatment and lead to a high CR rate.

Once radiological CR is achieved by TACE, its quality may be important to maintain durable local control. Previous studies have suggested that the specific value of iodized oil accumulation is associated with treatment response after TACE, and that the degree of iodized oil accumulation is an important factor for predicting tumor response in post-TACE evaluation [23,24,25,26]. Park et al. reported that the threshold value of >460 HU for iodized oil accumulation was highly sensitive and specific for pathological CR [12]. In the present study, the mean HU value of iodized oil accumulation after subsegmental B-TACE was 592 ± 1888 HU (range, 243–980 HU), which was higher than those values observed in the previous studies predicting pathological CR.

HCC often has microsatellite lesions that cannot be diagnosed by imaging modalities, and LR may occur from these untreated microsatellite lesions [27,28]. Previous studies reported that the LR rates of HCC with sufficient visualization of the peritumoral portal vein during C-TACE were significantly lower than those of HCCs with slight or no portal vein visualization [29,30,31]. Although iodized oil accumulation in the peritumoral liver parenchyma on CT is frequently identified immediately after C-TACE, it might frequently overestimate the embolized area because washout of iodized oil in the peritumoral liver parenchyma is frequently identified at the next follow-up CT [6]. This may be due to the fact that visualization of only the peritumoral portal vein during C-TACE does not always imply the successful embolization of the peritumoral liver parenchyma. However, in B-TACE, more lipiodol emulsion can be accumulated into the peritumoral portal vein, which has higher potential for peritumoral parenchymal necrosis [29]. Therefore, necrosis of the peritumoral liver parenchyma may be important to achieve successful control of possible microsatellite nodules around the tumor.

A novel finding of the present study was that oily subsegmentectomy including radiological CR in the HCC and peritumoral parenchymal necrosis was obtained in 28 (38.9%) of 72 HCCs at the first follow-up CT with a mean of 40 days (range, 17–63 days) after subsegmental B-TACE. Prominent visualization of the peritumoral portal vein during subsegmental B-TACE may suggest adequate blockage of both arterial and portal supplies to the tumor and peritumoral liver parenchyma, which can potentially induce oily subsegmentectomy. Moreover, regardless of tumor marker levels at pre-B-TACE, the tumor markers were normalized at the first follow-up date in all HCCs with oily subsegmentectomy. Additionally, HCCs with oily subsegmentectomy did not demonstrate any LR during the follow-up period. Thus, oily subsegmentectomy obtained by subsegmental B-TACE can potentially have a sufficient therapeutic effect on HCC. Therefore, oily subsegmentectomy could be used as an index when selecting a follow-up treatment strategy after B-TACE.

In the present study, the factors associated with a lower LR rate were age ≥65 years and peripheral location of the HCC. Granito A. et al. reported that posttreatment transient transaminase elevation was predictive of the objective response to superselective C-TACE, and the optimal cutoff points in predicting the CR were a 52% alanine aminotransferase and a 46% aspartate aminotransferase increase after C-TACE compared to the pretreatment values [32]. In the present study, we also found that oily subsegmentectomy was significantly associated with peripherally located HCC (*p* = 0.008). TACE is often performed to treat central HCCs. However, controlling these HCCs with superselective TACE can be difficult owing to multiple small feeding arteries branching directly from the trunk artery. In centrally located HCC, a greater volume of the peripheral liver parenchyma can be influenced by iodized oil when B-TACE is performed via the proximal trunk artery. Therefore, accumulating sufficient iodized oil in the HCC and peritumoral portal veins to obtain oily subsegmentectomy is difficult. According to our results, the best indication for subsegmental B-TACE is peripherally located HCC(s).

In the present study, the median PRS rate was 35.9 months (range, 29–43 months) and the cumulative PRS rates were 92.3%, 84.6%, 76.9%, 44.8%, and 44.8% at 6, 12, 24, 36, and 48 months, respectively. Facciorusso A. et al. reported that the median PRS rate after percutaneous radiofrequency ablation was 22 months [33]. However, comparison with this report is difficult and probably not meaningful because the patient groups may be different and a different treatment method was used.

Regarding safety, no studies have reported a significant difference in the rate of complications between procedures performed with and without balloon occlusion [15,16,17,18,19,20,21]. B-TACE has been demonstrated to be safe, with an adverse event rate equivalent to that of C-TACE. In the present study, asymptomatic ischemic cholangiopathy was noted in one patient. In addition, all the patients experienced increased biochemical marker levels, including AST, ALT, and total bilirubin, immediately after subsegmental B-TACE, and 56.7% of the patients experienced mild fever, nausea, and/or pain defined as post-embolization syndrome. However, the biochemical markers at the first follow-up had returned to baseline levels observed before B-TACE. In the present study, we did not observe an extensive area of parenchymal necrosis, and the range of peritumoral necrosis was always confined within 3 cm around the HCC.

The present study has several limitations. First, this retrospective single-institution study had a relatively small number of patients, which may cause bias since only patients eligible for RFA and SR were included. Hence, all the patients had preserved liver function and underwent selective B-TACE. Therefore, the findings of the present study might not be applicable to patients with advanced liver disease and multiple HCCs requiring extensive TACE. Second, the type and concentration of the chemotherapeutic agents and the volume of injected lipiodol differed between cases, which might have potentially affected the treatment outcomes. Third, oily subsegmentectomy was not achievable in all the patients after successful subsegmental B-TACE, and the exact reason is unknown. Fourth, the range of the peritumoral parenchymal necrosis cannot be controlled intentionally in some HCCs. Finally, the interval between B-TACE and the first follow-up CT varied among cases, and the HU of lipiodol accumulation after B-TACE might differ according to the timing of the first follow-up CT. However, the interval was fairly short (<2 months) in most patients (except one patient).

## 5. Conclusions

Subsegmental B-TACE can be an effective method with a high initial CR rate and low LR incidence. Oily subsegmentectomy can be considered an index of successful treatment because it did not demonstrate any LR.

## Figures and Tables

**Figure 1 cancers-15-04991-f001:**
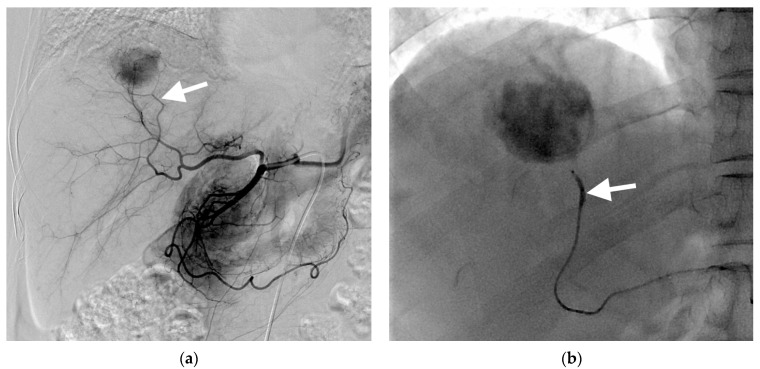
Images of a 67-year-old woman with hepatocellular carcinoma treated with subsegmental balloon-occluded transcatheter arterial chemoembolization (B-TACE). (**a**) Hepatic arteriography demonstrating a hypervascular tumor in the right posterior superior segment. The tumor is wholly supplied by the posterior superior sectional artery (arrow). (**b**) A microballoon catheter was successfully placed in the feeding artery. A single fluoroscopic image presenting subsegmental B-TACE, consisting of lipiodol–cisplatin injection during microballoon inflation (arrow). (**c**) A single fluoroscopic image demonstrating dense lipiodol accumulation within the tumor (asterisk) and peritumoral portal veins. (**d**) A single fluoroscopic image after subsegmental B-TACE demonstrating dense lipiodol accumulation within the tumor (asterisk) and peritumoral portal veins and liver parenchyma (arrowheads). (**e**) Final angiography presenting successful embolization of the feeding artery without residual tumor staining. (**f**) A multiphase coronal computed tomography in the portal phase image 1 month after B-TACE, demonstrating oily subsegmentectomy with dense lipiodol accumulation within the tumor (asterisk) and non-enhancing, hypoattenuating peritumoral liver parenchyma, indicating peritumoral parenchymal necrosis (arrowheads). The tumor had 967 HU, and the maximum length of peritumoral necrosis from the tumor was 15.6 mm.

**Figure 2 cancers-15-04991-f002:**
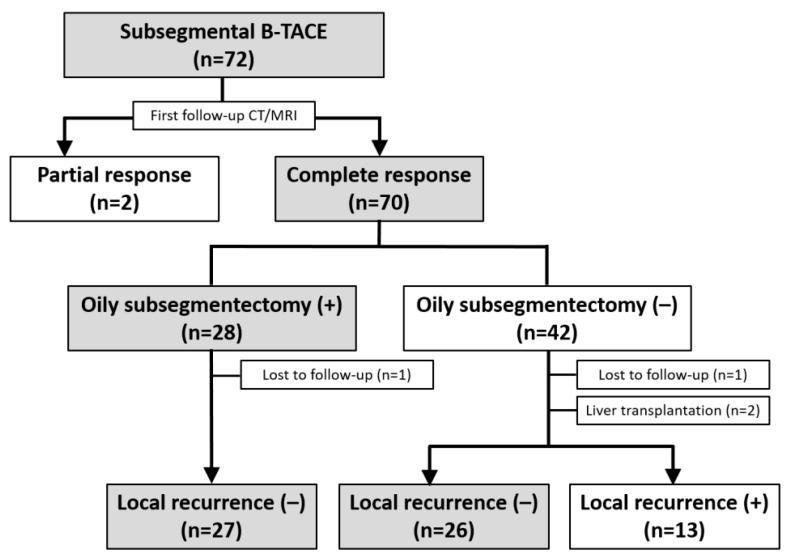
Flow diagram of the treatment outcomes after subsegmental balloon-occluded transcatheter arterial chemoembolization.

**Figure 3 cancers-15-04991-f003:**
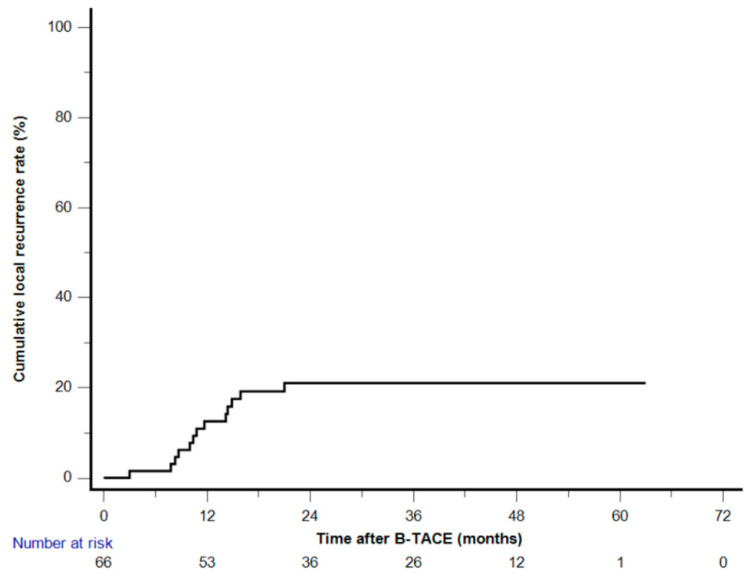
Kaplan–Meier curve of the cumulative local recurrence after balloon-occluded transcatheter arterial chemoembolization.

**Figure 4 cancers-15-04991-f004:**
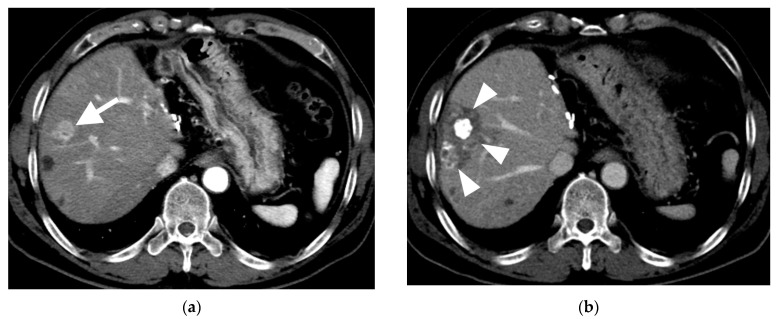
Computed tomography (CT) images of hepatocellular carcinoma treated with subsegmental balloon-occluded transcatheter arterial chemoembolization (B-TACE) in a 60-year-old man. (**a**) A multiphase axial CT in the arterial phase demonstrating an enhancing tumor with a maximal diameter of 2 cm (arrow). (**b**) A multiphase axial CT in the portal phase image 1 month after B-TACE, indicating oily subsegmentectomy (arrowheads). (**c**) A multiphase axial CT in the portal phase image 1 month after B-TACE, demonstrating a reduction in the oily subsegmentectomy area and tumor size (17 mm). (**d**) A multiphase axial CT in the portal phase image 4 years after B-TACE, demonstrating a reduction in tumor size of 4 mm without a viable tumor.

**Figure 5 cancers-15-04991-f005:**
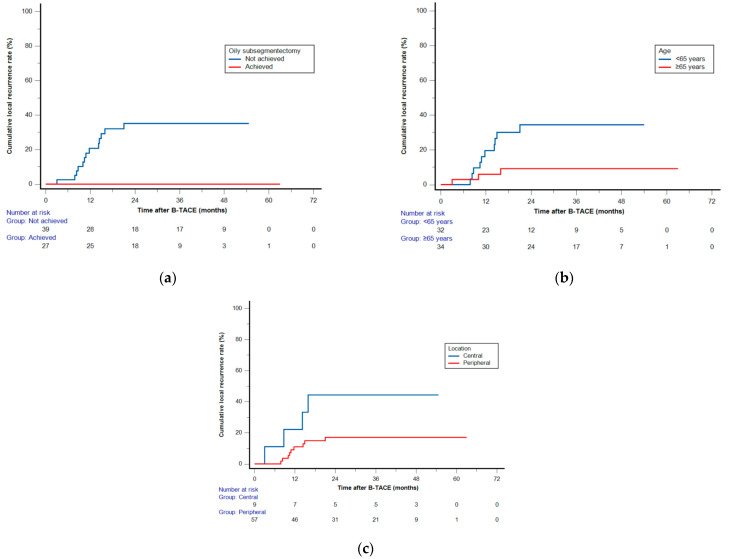
Log-rank analysis (**a**) and multiple Cox proportional hazard regression analyses (**b**,**c**) of cumulative local recurrence (LR) rates after balloon-occluded transcatheter arterial chemoembolization. Kaplan–Meier curves demonstrating the cumulative LR rates according to (**a**) oily subsegmentectomy, (**b**) age, and (**c**) hepatocellular carcinoma location.

**Table 1 cancers-15-04991-t001:** Baseline characteristics of the 60 study patients.

Characteristics	Value
Age (years)	63.5 ± 10 (39–82)
Sex, male	44 (73.3%)
Hepatitis B virus infection	41 (68.3%)
Hepatitis C virus infection	3 (5%)
Alcohol	16 (26.7%)
BCLC stage A	50 (83.3%)
BCLC stage B	10 (16.7%)
Child–Pugh class A	57 (95%)
Child–Pugh class B	3 (5%)
MELD	7.9 ± 1.3 (6–11)
Aspartate aminotransferase (U/L)	36 ± 18 (12–119)
Alanine aminotransferase (U/L)	33 ± 25 (12–165)
Total bilirubin (mg/dL)	0.8 ± 0.4 (0.2–1.8)
α-Fetoprotein (ng/mL) ^1^	829 ± 3313 (2.3–8899.6)
PIVKA-II (mAU/mL)	41 ± 193 (1.7–1425.4)
HCC size (mm)	31 ± 10.6 (10–50)
Single HCC	49 (81.7%)
Two HCCs	10 (16.7%)
Three HCCs	1 (1.7%)
No previous treatment	46 (76.7%)
Previous surgical resection	9 (15%)
Previous RFA	5 (8.3%)

^1^ Data available for 53 patients.

**Table 2 cancers-15-04991-t002:** Results of the univariate and multiple Cox proportional hazard model for predicting the factors associated with local recurrence.

Factors	Univariate Analyses		Multiple Analyses	
	Hazard Ratio (95% CI)	*p*-Value	Hazard Ratio (95% CI)	*p*-Value
Male sex	0.637 (0.235–1.724)	0.375		
Age ≥ 65 years	0.243 (0.074–0.804)	0.021	0.124 (0.037–0.412)	<0.001
Viral liver cirrhosis	1.791 (0.455–7.048)	0.404		
Child–Pugh B	Inestimable *	0.600		
MELD ≥ 8	1.604 (0.600–4.291)	0.347		
α-Fetoprotein > 830 ng/dL	0.554 (0.091–3.364)	0.521		
Previous treatments	0.321(0.048–2.148)	0.241		
HCC size > 3 cm	0.475 (0.155–1.454)	0.192	0.416 (0.138–1.256)	0.120
Multiple HCCs	1.153 (0.447–2.975)	0.768		
Peripheral location	0.319 (0.101–1.003)	0.051	0.112 (0.046–0.272)	<0.001
Oily subsegmentectomy	Inestimable ^†^	0.001 ^‡^		

* Hazard ratio was inestimable since neither patient with Child–Pugh B class experienced local recurrence. ^†^ Hazard ratio was inestimable since no HCCs with oily subsegmentectomy demonstrated local recurrence. ^‡^ Since the Cox regression analysis was not applicable, log-rank test was used to calculate *p*-value.

## Data Availability

The data presented in this study are available upon justified request.

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
