# Peer review of "Local Recurrence following Radiological Complete Response in Patients Treated with Subsegmental Balloon-Occluded Transcatheter Arterial Chemoembolization for Hepatocellular Carcinoma"

_cancers, 2023, doi:10.3390/cancers15204991_

Round 1

Reviewer 1 Report

Very interesting study. The retrospective design and the limited sample size represent major limitations and should be discussed as such in the discussion.

Did the authors check the assumptions for proportional regression analysis? (homoschedasticity, normal distribution, collinearity....)

The authors should comment on the concept of post-recurrence survival after loco-regional treatments (cite the series PMID: 25085684)

Among the potential predictors of response and local recurrence, the authors could test also the post-TACE hypertransaminasemia. At least comment this concept in the discussion citing the recent series (PMID: 34683182)

Did any patients undergo radioembolization after TACE recurrence?

Author Response

Reviewer #1

1. Very interesting study. The retrospective design and the limited sample size represent major limitations and should be discussed as such in the discussion.

Response) Thank you for your valuable comments. We already mentioned these limitations in the DISCUSSION section.

2. Did the authors check the assumptions for proportional regression analysis? (homoschedasticity, normal distribution, collinearity....)

Response) Thank you for your valuable comments. All the variables did not show any significant correlation. Multivariate Cox’s regression analysis could adjust the potential variables.

3. The authors should comment on the concept of post-recurrence survival after loco-regional treatments (cite the series PMID: 25085684)

Response) Thank you for your valuable comments. We added overall survival and post-recurrence survival (PRS) at the end of RESULT as follows. “Following B-TACE, 12 (22.2%) of 54 patients died, whereas 42 (77.8%) remained alive. The causes of death were liver failure in 7 (58.3%) patients, tumor progression in 3 (25%), and events other than liver decompensation or tumor progression in 2 (16.7%). The cumulative survival rates were 100%, 94.4%, 88.9%, 84.9%, and 78.5% at 6, 12, 24, 36, and 48 months, respectively. Median PRS rate was 35.9 months (range, 29–43 months) and cumulative PRS rates were 92.3%, 84.6%, 76.9%, 44.8%, and 44.8% at 6, 12, 24, 36, and 48 months, respectively.” In the DISCUSSION, we also mentioned about PRS.

4. Among the potential predictors of response and local recurrence, the authors could test also the post-TACE hypertransaminasemia. At least comment this concept in the discussion citing the recent series (PMID: 34683182) TRANS-TACE: Prognostic Role of the Transient Hypertransaminasemia after Conventional Chemoembolization for Hepatocellular Carcinoma

Response) Thank you for your valuable comments. We added the important conclusions of the reference in the DISCUSSION.

5. Did any patients undergo radioembolization after TACE recurrence?

Response) No patients with local recurrence undergo any radioembolization after B-TACE recurrence. All 13 patients with a local recurrence were suitable for a second conventional TACE treatment. We added it in the local recurrence of the RESULT section.

Reviewer 2 Report

 The main question of this study was to determine the local recurrence (LR) rate and identify factors associated with LR in patients who achieve a radiological complete response (CR) after undergoing 23 balloon-occluded transcatheter arterial chemoembolization (B-TACE) for hepatocellular carcinoma 24 (HCC). The planning and design of the study is good, the patient number is enough, the results are promising. The safety data are similar like in previous papers. The novel finding of the present study was that oily subsegmentectomy including ra-275 diological CR in the HCC and peritumoral parenchymal necrosis was obtained in 28 276 (38.9%) of 72 HCCs at the first follow-up CT with a mean of 40 days (range, 17–63 days) 277 after subsegmental B-TACE. The references could be more  uptodate, I recommend to integrate the reviews of the last years. The Figure 5. could be a little bit bigger. Explanation of the study limitation is satisfying. The conclusion part is straightforward.

Author Response

The main question of this study was to determine the local recurrence (LR) rate and identify factors associated with LR in patients who achieve a radiological complete response (CR) after undergoing 23 balloon-occluded transcatheter arterial chemoembolization (B-TACE) for hepatocellular carcinoma 24 (HCC). The planning and design of the study is good, the patient number is enough, the results are promising. The safety data are similar like in previous papers. The novel finding of the present study was that oily subsegmentectomy including ra-275 diological CR in the HCC and peritumoral parenchymal necrosis was obtained in 28 276 (38.9%) of 72 HCCs at the first follow-up CT with a mean of 40 days (range, 17–63 days) 277 after subsegmental B-TACE. The references could be more  uptodate, I recommend to integrate the reviews of the last years. The Figure 5. could be a little bit bigger. Explanation of the study limitation is satisfying. The conclusion part is straightforward.

Response) Thank you for your valuable comments. Most of the recent B-TACE papers were included in the reference listt. So, we added one reference (reference number 21) published in 2022. And we expanded the size of three figures in the Figure 5.

Round 2

Reviewer 1 Report

The revised version of the manuscript is OK. Thank you!